# Tunable Broadband-Narrowband and Dual-Broadband Terahertz Absorber Based on a Hybrid Metamaterial Vanadium Dioxide and Graphene

**DOI:** 10.3390/mi14010201

**Published:** 2023-01-13

**Authors:** Jing Li, Yanfei Liu, Yu Chen, Wenqing Chen, Honglei Guo, Qiannan Wu, Mengwei Li

**Affiliations:** 1School of Instrument and Electronics, North University of China, Taiyuan 030051, China; 2School of Instrument and Intelligent Future Technology, North University of China, Taiyuan 030051, China; 3Academy for Advanced Interdisciplinary Research, North University of China, Taiyuan 030051, China; 4Center for Microsystem Intergration, North University of China, Taiyuan 030051, China; 5School of Semiconductors and Physics, North University of China, Taiyuan 030051, China; 6Key Laboratory of Dynamic Measurement Technology, North University of China, Taiyuan 030051, China

**Keywords:** terahertz, metamaterial absorber, vanadium dioxide, graphene

## Abstract

We propose a functionally tunable terahertz (THz) metamaterial absorber, which has the switching performance between broadband-narrowband and dual-broadband near-perfect absorption due to the phase transition of Vanadium dioxide (VO_2_) and the tunable electrical property of graphene. The switching absorption properties are verified by computer simulation technology (CST) microwave study. The simulation results show that when VO_2_ is in the metallic phase, over 90% broadband absorption is realized in the 3.85–6.32 THz range. When the VO_2_ is in the insulating phase, the absorber shows quadruple narrowband absorption. By changing the Fermi level of graphene and the conductivity of VO_2_, the low-frequency broadband of 3.85–6.32 THz can be switched to the high-frequency broadband of 6.92–8.92 THz, and the absorber can be switched from a quadruple narrowband to a nearly singlefold narrowband. In addition, the proposed absorber is insensitive to polarization due to its symmetry and wide incident angle. The design may have potential applications in the THz range, such as switches, electromagnetic shielding, cloaking objects, filtering, sensing, and so on.

## 1. Introduction

Terahertz (THz) refers to electromagnetic waves with frequencies ranging from 0.1 THz to 10 THz and wavelengths ranging from 30 μm to 3 mm. THz technology has advantages over X-rays due to its high frequency, short pulse, strong penetration, low energy, and little damage to substances and human body. Therefore, it has broad prospects in space exploration, medical imaging, security inspection, broadband communication, and so on. In recent decades, with the rapid development of these fields, THz devices based on THz technology have also received extensive attention [1,2,3], such as filters, reflectors, absorbers, lens, etc. Among them, the THz metamaterial perfect absorber, which can achieve the effect of complete absorption without reflecting or transmitting the incident electromagnetic wave in a certain frequency point or frequency band, has aroused great interest in the academic circle.

In recent years, many THz metamaterial absorbers have been reported. In 2020, Zhong et al. proposed a THz absorber based on semi-elliptical media and graphene that not only achieves ultra-fast band absorption, but is also insensitive to light polarization in the 50° range of incident angles [4]. In 2021, Feng et al. proposed a polarimetric insensitive absorber based on a cross-shaped graphene sheet that can perform wide-band absorption in the frequency range of 1.23–1.68 THz [5]. In 2022, Pouria Zamzam et al. proposed a dual to four-band polarization-insensitive absorber based on additional graphene layers. The absorber has the function of wide incidence angle and polarization insensitive absorption [6]. In 2019, Wang et al. proposed a terahertz metamaterial device based on a double-cycle structure of vanadium dioxide [7]. By studying the effect of incident angle on absorbance, its broadband absorption performance over a ±50° field-of-view for the frequency band ranges from 0.61 to 1.36 THz. However, for these THz metamaterial absorbers, there are still some problems, such as structural obsolescence, limited frequency or wavelength, low absorption efficiency, unadjustable absorption performance, and insensitive polarization, which greatly limit their further practical application. Therefore, how to realize the high performance tunable THz metamaterial absorber has become an important research direction of THz science and technology.

To solve the above problems, we design a multi-tuned hybrid metamaterial THz absorber based on vanadium dioxide (VO_2_) and graphene. It is noted that graphene is a 2D material with a single layer of carbon atoms [8,9,10], whose Fermi energy level (EF) can be continuously changed by controlling chemical doping or gate voltage, providing an excellent dynamic regulation function in the THz absorber. In addition, VO_2_ is also an excellent tuning material, whose dielectric constant can be reversed from the insulating phase to the metallic phase [11,12,13], and the phase transition time can be completed in picoseconds. The fast and dynamic absorption tuning can be demonstrated using a combination of graphene and VO_2_. In this letter, the absorption mechanism is explained by using electric field distribution and impedance matching theory, and the polarization and incident absorption characteristics of TE and TM at different incident angles are studied. The advantages of this absorber are novel structure, wide tuning range, and fast response time, which is expected to be widely used in THz systems.

## 2. Structure Design and Method

A 3D schematic of the designed broadband-narrowband and dual-broadband terahertz metamaterial absorbers and a top view of the *x*-*y* plane are shown in Figure 1a,b. The absorbers consist of four layers, from bottom to top, a metal mirror layer, an insulator layer, a single graphene layer, and a VO_2_-gold (Au) pattern layer on top. As shown in the Figure 1, the structure of the absorber unit is arranged periodically in the *x* and *y* directions with *P* = 40 µm. The thickness of the bottom gold is *T*_1_ = 200 nm. For the insulation layer, the thickness is *T*_2_ = 9 µm. We assume that the material is lossless and the relative dielectric constant is 1.96 [14]. The VO_2_-Au pattern layer on the top layer is composed of an Au-ring with a cross opening and four T-shaped VO_2_ resonators. Among them, the thickness of VO_2_ layer *T*_3_ = 200 nm, the geometric parameters are *L*_1_ =17.5 µm, *L*_2_ = 5 µm, *L*_3_ = 4.5 µm, *L*_4_ = 6 µm, *W*= 2.5 µm; the optimization parameters of the cross-shaped open metal ring located in the center are *R*_1_ = 4 µm, *R*_2_ = 8 µm. *g* = 0.625 µm; and the thickness is consistent with *T*_3_, where the conductivity of gold, *σ* gold = 4.09 × 10^7^ S/m [15]. The metal as a reflective layer can ensure that almost all incident THz waves are reflected, thus inhibiting transmission.

The interaction between electromagnetic waves and graphene can be explained by solving Maxwell’s equation. All the calculations were carried out by using Computer Simulation Technology (CST) microwave studio, and the 3D numerical results were obtained. In the simulation, graphene is modeled as an infinitely thin surface characterized by surface conductivity *σ_gra_*, Therefore, the surface conductivity of graphene can be described by the following formulas (i.e., Kubo formula) [16]:(1)$$\sigma_{gra}=\sigma_{intra}+\sigma_{inter},$$
(2)σintra=2e2kBTπh2iω+iτIn2coshEf2kBT,
(3)$$\sigma_{inter}=\frac{e^{2}}{4h^{2}}\left[\frac{1}{2}+\frac{1}{\pi }arctan\left(\frac{h\omega -2E_{f}}{2}\right)-\frac{i}{2\pi }In\frac{{\left(h\omega +2E_{f}\right)}^{2}}{{\left(h\omega +2E_{f}\right)}^{2}+4{\left(k_{B}T\right)}^{2}}\right],$$
where *T*, $k_{B}$, *e*, τ, ω, *h*, and *E_f_* are the ambient temperature (*T* = 293 K), Boltzmann constant, electron charge, carrier relaxation time, angular frequency of incident light, reduced Planck constant, and Fermi level of graphene, respectively. In the THz range, the interband contribution can be ignored due to the intraband contribution dominates. Accordingly, the above Kubo formula can be simplified into a Drude-like model [17]:(4)σgra=e2Efπh2iω+iτ,

It can be seen from Equation (4) that the surface conductivity of graphene is not only related to the angular frequency and relaxation time of the incident electromagnetic wave, but also to the Fermi level. Therefore, we adjust the surface conductivity of graphene by applying a bias voltage to it to change the Fermi level of graphene. In this paper, the Fermi level is chosen as *E_f_* = 0 eV and carrier relaxation time τ = 0.1 ps for broadband and narrowband absorption.

The Drude model also allows us to perform optical characterization of VO_2_ in the THz range, where the dielectric constant is expressed as [18]:(5)$$\varepsilon {\left(\omega \right)}_{VO_{2}}=\varepsilon_{\infty }-\frac{\omega_{p^{2}}^{2}}{\omega {\left(\omega +i\gamma_{2}\right)}^{}},$$

In Equation (5), where ω is the frequency of an incident electromagnetic wave, $\varepsilon_{\infty }$ is the high frequency relative dielectric constant of VO_2_, which is 12, $\gamma_{2}$ is the collision frequency, which is 5.57 × 10^13^, where $\omega_{p^{2}}$ is the plasma frequency related to the conductivity of VO_2_, which can be approximated as:(6)$$\omega_{p^{2}}^{2}=\frac{\sigma }{\sigma_{0}}\omega_{p^{0}}^{2},$$

In Equation (6), *σ*_0_ = 3 × 10^5^, $\omega_{p^{2}}^{2}$ = 1.4 × 10^5^, *σ* is the VO_2_ conductivity. When the temperature changes, VO_2_ can be converted from an insulating phase to a metallic phase [19]. Figure 2a shows the relationship between the conductivity of VO_2_ and the ambient temperature. At room temperature, VO_2_ presents an insulating phase with a conductivity of 20 S/m. When the temperature reaches the phase transition temperature, VO_2_ is in the metallic phase and the conductivity is 2 × 10^5^ S/m. Figure 3b,c shows the relationship between the real parts (Re (*ε*))and imaginary parts (Im (*ε*)) of the relative dielectric constant of VO_2_ under different phase transitions, respectively. It is noted that the Re (*ε*) is the symbols from positive to negative. When VO_2_ is exchanged from the insulating phase (10 S/m) to the metallic phase (2 × 10^5^ S/m), Im (*ε*) completely increasing by degrees.

In this paper, the performance of the THz metamaterial absorber was tested using the CST microwave studio. In the simulation, periodic boundary conditions were set in the *x* and *y* directions. The Floquet port was set in the *z* direction. The narrowband and broadband absorption spectra of the THz metamaterial absorber in the insulating phase and metallic phase of VO_2_ were obtained by simulation. In the simulation, the absorption rate can be expressed as:*A* = 1 − *R* − *T,*(7)

Among them, *A*, *R*, and *T*, respectively for the absorptivity, reflectance, and transmittance, *R* = |*S*_11_|^2^, *T* = |*S*_21_|^2^. |*S*_11_| and |*S*_21_| represent reflection coefficient and transmission coefficient, respectively. Since the bottom metal can completely reflect electromagnetic waves, the transmittance *T* is 0, which can be simplified as *A* = 1 − *R* [20].

## 3. Results and Discussion

Figure 3 shows the absorption spectra obtained using CST. The broadband red and narrowband blue lines show the absorption curves of the VO_2_ in the metallic phase (*σ* = 2 × 10^5^ S/m) and the insulating phase (*σ* = 20 S/m), respectively, when the Fermi level of graphene is 0 eV. It was observed that more than 90% broadband absorption was achieved in the range 3.85–6.32 THz (red line) when the VO_2_ was in the metallic phase. When the VO_2_ is in the insulation phase, the four narrowband absorption peaks (blue line) appear at 6.4 THz, 7.32 THz, 8.49 THz, and 9.78 THz, and the absorption amplitudes of the four narrowband absorption peaks are all greater than 70%, which are 95.2%, 97%, 89.1%, and 73.3%, respectively. Among them, the broadband and narrowband performance parameters of graphene and VO_2_ corresponding to the absorber under different conditions are shown in Table 1. In addition, the TE and TM modes have the same absorption due to the symmetry of the structure. Therefore, by controlling the phase transition of the VO_2_, the function of the terahertz absorber can be flexibly switched between broadband and four narrowband.

In order to explain the absorption mechanism of the absorber, Figure 4a–f shows the electric field distribution of broadband absorption and narrowband absorption at different resonant frequencies of VO_2_ in the metallic phase and insulating phase. The electric field of the incident wave is polarized along the *y*-axis, and the color plot represents the intensity of the field. For a broadband absorber, Figure 4a,b shows the electric field distribution at the frequency of two absorption peaks of 4.74 THz and 8.78 THz. At low resonant frequency, the resonance between the VO_2_ material and the insulator dielectric layer will excite the surface isobaric body, resulting in higher electric field intensity around T-VO_2_ and enhanced absorption performance of the absorber. With the increase of frequency, due to the resonance effect between the cross-shaped open metal ring and the T-shaped VO_2_ at 8.78 THz, the electric field in the high resonant frequency of the broadband absorber is mainly located outside the metal ring and the VO_2_. In addition, in order to understand the physical mechanism of the four narrowband absorption peaks of the absorber when VO_2_ is the insulating phase, as shown in Figure 4c–f, we study the electric field distribution at the frequencies of 6.39 THz, 7.32 THz, 8.49 THz, and 9.78 THz, where VO_2_ is equivalent to the dielectric. In Figure 4c, the electric field at 6.39 THz is mainly concentrated in the gap between the single graphene layer and the cross-shaped open metal ring. With the increase of frequency, the electric field in the gap between the single graphene layer and the cross-shaped open metal ring increases at 7.32 THz and gradually expands towards the edge of the ring (Figure 4d). However, as shown in Figure 4e,f, the electric field gradually weakens at 8.49 THz and 9.78 THz. This may be due to the fundamental resonance mode of the graphene structure (electric dipole resonance).

In order to understand the underlying physical mechanism of the absorption phenomenon, impedance matching theory is used. Among them, the influence of different electrical conductivity on absorption spectra during the phase transition of VO_2_ is shown in Figure 5a. The results show that when VO_2_ is in the metallic phase (*σ* = 2 × 10^5^ S/m), the absorbers have remarkable performance in broadband and narrowband, respectively.

The absorption rate of the absorber can be expressed as:(8)$$A=1-R=1-{\left|\frac{Z-Z_{0}}{Z+Z_{0}}\right|}^{2}=1-{\left|\frac{Z_{r}-1}{Z_{r}+1}\right|}^{2}$$
(9)$$Z_{r}=\pm \sqrt{\frac{{\left(1+S_{11}\right)}^{2}-S_{21}^{2}}{{\left(1-S_{11}\right)}^{2}-S_{21}^{2}}},$$
where *S*_11_, *S*_21_, *Z*, and *Z_0_* are *S* parameters, effective impedance, and free space impedance, respectively. *Z_r_* = *Z*/*Z*_0_ represents the relative impedance between the absorber and the free space. The necessary condition for a metamaterial absorber to achieve perfect absorption is that the equivalent impedance of the metamaterial absorber matches the impedance of the free space, so that the incident electromagnetic wave enters the absorber to the maximum extent, and then the reflection reaches the minimum, so as to realize the ultra-broadband absorption characteristics. Among them, the real part and imaginary parts of the relative impedance (*Z_r_*) of the absorber at different conductivity of VO_2_ are shown in Figure 5b,c. When VO_2_ is in a metallic phase, the real part approaches 1 and the imaginary part approaches 0 (black dashed line) in the frequency range of 3.85–6.32 THz, which means that the impedance of the proposed absorber almost matches the free space. At this time, the reflection of the absorber structure on the incident electromagnetic wave is almost zero, and the maximum loss of the incident terahertz wave is inside the insulation layer, achieving nearly perfect absorption.

Figure 6 shows the tunable function of the absorber in broadband and narrowband. By changing the Fermi level of graphene, the working bandwidth and intensity of narrowband absorption and broadband absorption can be dynamically adjusted. Figure 6a shows that when VO_2_ is in the metallic phase and the Fermi level of graphene is 0 eV, the low-frequency broadband (solid black line) appears in the frequency range of 3.85–6.32 THz. As the Fermi level of graphene increases to 0.7 eV, the low-frequency broadband shows an obvious blue shift. The high-frequency broadband (solid red line) is displayed in the frequency range of 6.92–8.92 THz, and there has been a significant change in broadband absorption intensity. With the disappearance of the low-frequency broadband with an absorption rate of more than 90%, the absorption rate gradually decreases, and when the high-frequency broadband appears, the absorption rate is tuned to more than 90% in the range of 6.92–8.92 THz. By keeping the change in the graphene Fermi energy level constant, Figure 6b shows that the absorber can be switched from a quadruple narrowband (solid black line) to something close singlefold narrowband (solid red line) when the VO_2_ is set to an insulating phase and the Fermi level of graphene is 0 eV. Among them, the broadband and narrowband performance parameters of graphene and VO_2_ corresponding to the absorber under different conditions are shown in Table 1. This indicates that the proposed absorbers have potential applications in optical switching.

Finally, in order to study the absorption effect of the absorber under oblique incidence, the broadband and narrowband absorption spectra with the polarization angle from 0° to 85° and incident angle from 0° to 85° with step size of 10° are simulated in the frequency range of 1–10 THz. Figure 7a,b and Figure 8a,b show the absorption spectra of broadband and narrowband absorbers under TM and TE polarized waves, respectively. Obviously, with the increase of the polarization angle, the broadband absorption rate and narrowband absorption rate did not change. Therefore, the proposed absorber is insensitive to the polarization angle. This may be due to the complete symmetry of the designed structure. In addition, the influence of different incidence angles on the absorption of TM and TE waves in broadband and narrowband is also studied. Figure 7c,d shows the absorption spectra of the broadband absorber at the incidence angles of TM and TE, respectively. As shown in Figure 7c, when the angle is lower than 55°, the absorption effect of 3.85–6.32 THz in the TM polarization state is better. However, when the incident angle exceeds 55°, the stable absorption of the broadband absorber is disturbed. The reason is that the tangential component of the electric field decreases with the increase of the incident angle. For TE incidence in Figure 7d, the absorption amplitudes can remain above 90% in the frequency range of 3.85–6.32 THz when the incidence angle reaches 20°. Figure 8c,d shows the narrowband absorption rates of TM and TE waves as functions of frequency and the incidence angle. For normal incidence, the absorptivity of the absorber has four absorption peaks at the frequencies of 6.4 THz, 7.32 THz, 8.49 THz, and 9.78 THz, and the absorption levels are about 95.2%, 97%, 89.1%, and 73.3%. For the TM polarization in Figure 8c, the number of absorption amplitudes increases as the incidence angle approaches 60°, and at the frequencies of 6.55 THz, 7.71 THz, 7.95 THz, 8.50 THz, and 9.78 THz, the absorption levels approach 96.4%, 99.6%, 94,5%, 95.5%, and 99.8%, respectively. As for TE incidence in Figure 8d, when the incidence angle is 30°, the number of absorption peaks increases gradually. When it is increased to 40°, five absorption peaks appear in the range of 6.86–8.77 THz, and the absorption amplitudes are all above 90%. With the increase of incidence angle, the absorption rate of the TE and TM wave at different polarization angles and incident angles decreases slowly, and the blue shift of frequency is strong.

At present, THz metamaterial absorbers based on VO_2_ and graphene have attracted more attention from researchers. To illustrate the advantages of the absorbers designed in this paper, we compare the tunable absorbers designed with other recently published absorbers in Table 2 and Table 3. In Table 2, we compare the broadband absorption characteristics with other published papers. Among them, the designed absorber has wide bandwidth and wide adjustability. In Table 3, we compare the narrowband characteristics of our absorbers with other devices. Our absorbers consist of four layers, while other absorbers have more layers, which simplify the manufacturing process. In addition, the device realizes the quadruple narrowband absorption under the condition of layer number optimization. It can also be observed from Figure 1 that the structure of the designed absorber is novel. Based on the above analysis, the advantages of the proposed terahertz metamaterial absorber have novel structure, wide adjustable range, and good performance.

## 4. Conclusions

In conclusion, a functionally tunable absorber based on VO_2_ and graphene is proposed, which can realize the switching performance of broadband-narrowband and double-bandwidth nearly perfect absorption by using the phase transition of VO_2_ and the electrical tunability property of graphene. When VO_2_ is in the metallic phase, the metamaterial device can be used as an adjustable broadband absorber, absorbing amplitudes of more than 90% in the range of 3.85–6.32 THz. Electrically controlled graphene Fermi energy levels enable double-broadband switching. In addition, the broadband absorber has good absorption performance at the TM incidence angle. Its absorption amplitudes are maintained above 90% in a wide range of the incidence angle up to 55°. When VO_2_ is in the insulating phase, the absorber can switch from broadband absorption to quadruple narrowbands absorption, and the maximum absorption amplitude of the four narrowbands is greater than 70%. At this time, with the change of the graphene Fermi energy level, the absorber can switch from a quadruple narrowband to a singlefold narrowband. In addition, the design is polarization independent. The proposed absorber provides a new method for the development of THz mirrors and absorption filters for broadband and multi-band applications. This research provides new ideas for the development of functional optics and energy harvesting devices.

## Figures and Tables

**Figure 1 micromachines-14-00201-f001:**
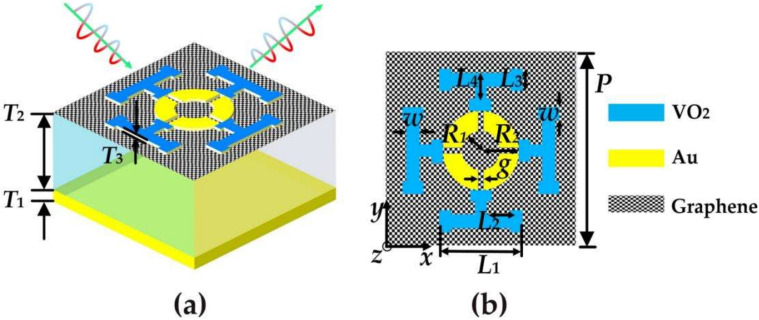
(**a**) 3D schematic of a terahertz metamaterial absorber; (**b**) top view of the *x*-*y* plane.

**Figure 2 micromachines-14-00201-f002:**
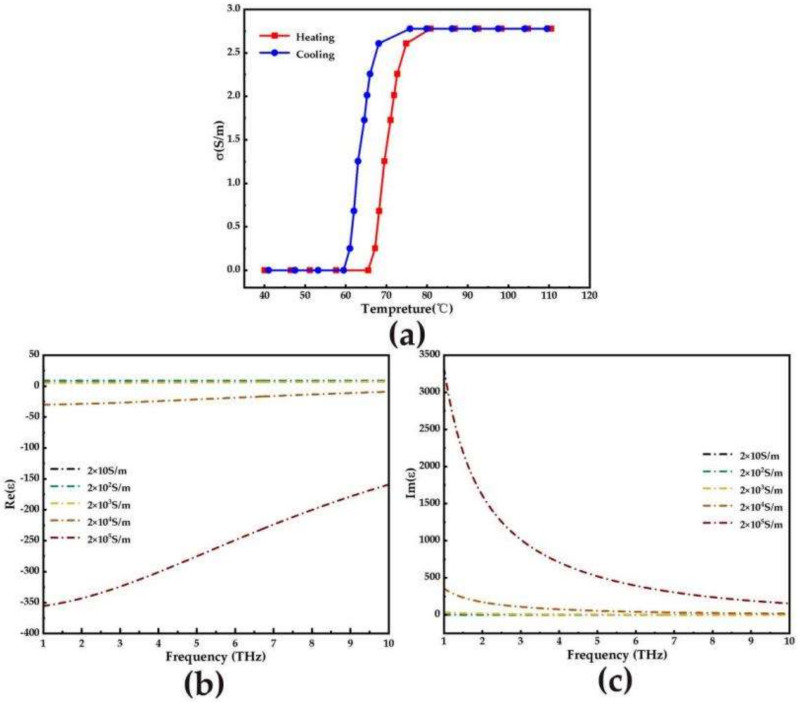
(**a**) The curves of the conductivity of the VO_2_ with ambient temperature; (**b**,**c**) the real and imaginary parts of the relative permittivity of VO_2_ at diverse conductivities.

**Figure 3 micromachines-14-00201-f003:**
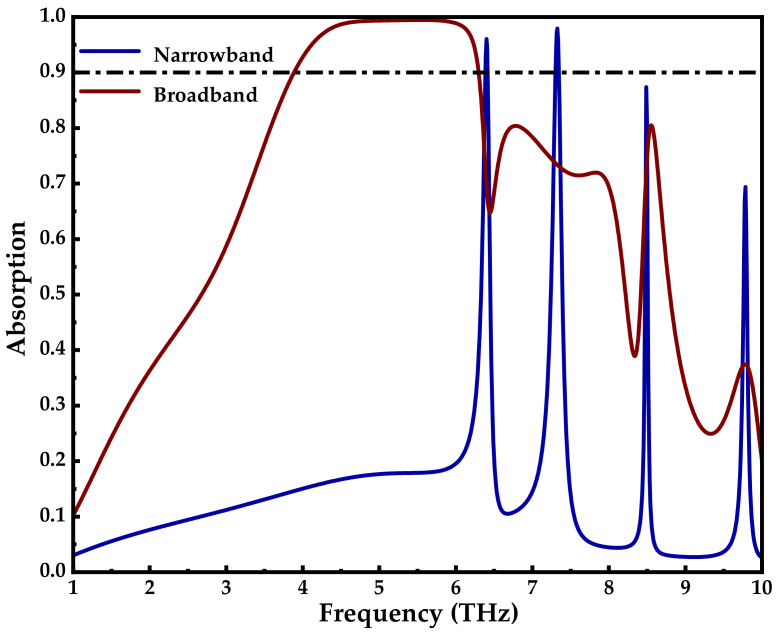
Broadband absorption curve of VO_2_ in metallic phase (red line) and narrowband absorption curve of VO_2_ in insulating phase (blue line).

**Figure 4 micromachines-14-00201-f004:**
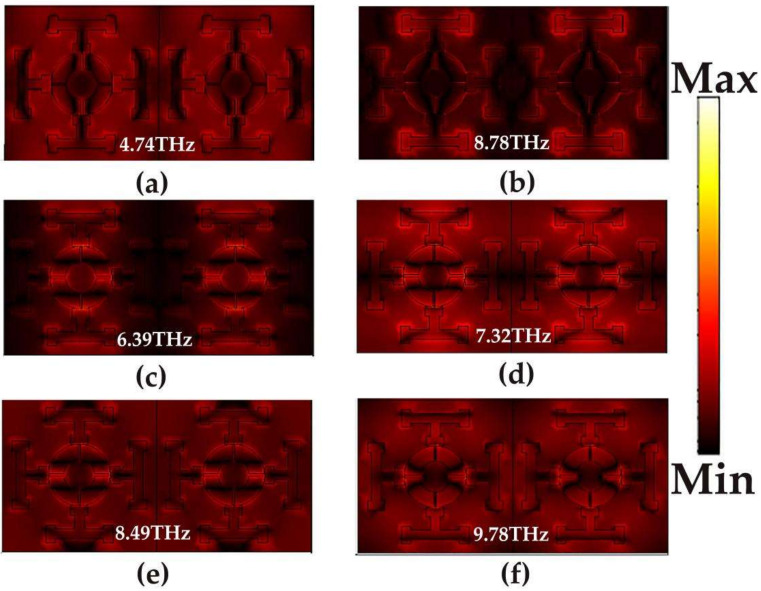
Electric field distribution of a broadband absorption and narrowband absorption: (**a**) 4.74 THz; (**b**) 8.78 THz; (**c**) 6.39 THz; (**d**) 7.32 THz; (**e**) 8.49 THz; (**f**) 9.78 THz.

**Figure 5 micromachines-14-00201-f005:**
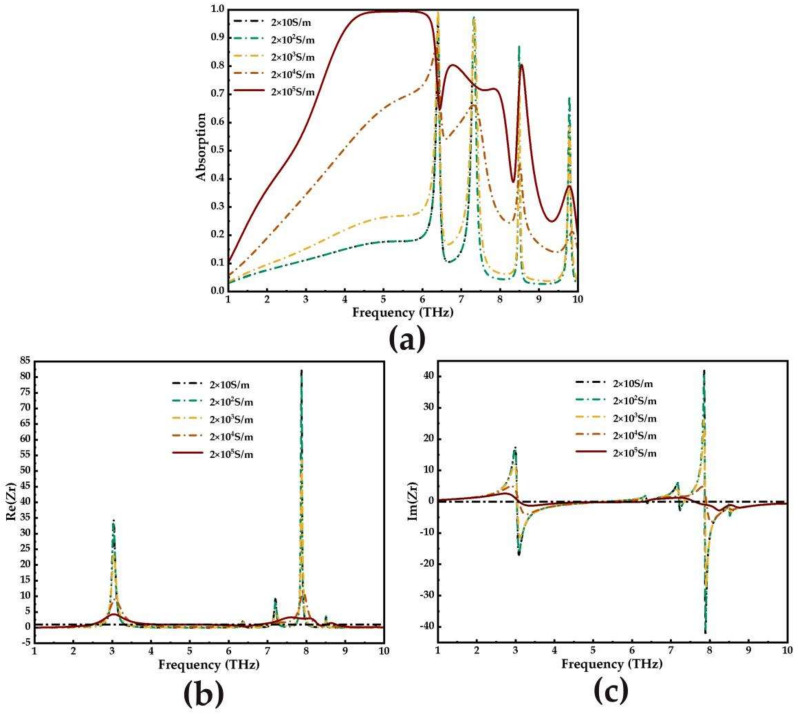
(**a**) Effect of different conductivity on absorption spectra during phase transition of VO_2_; (**b**) real parts and (**c**) imaginary parts of the relative impedance (*Zr*) with different conductivities of VO_2_.

**Figure 6 micromachines-14-00201-f006:**
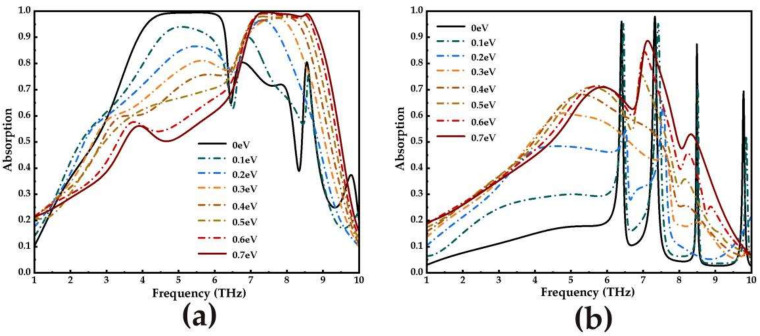
Under different Fermi level conditions of graphene: (**a**) broadband absorption spectrum; (**b**) narrowband absorption spectrum.

**Figure 7 micromachines-14-00201-f007:**
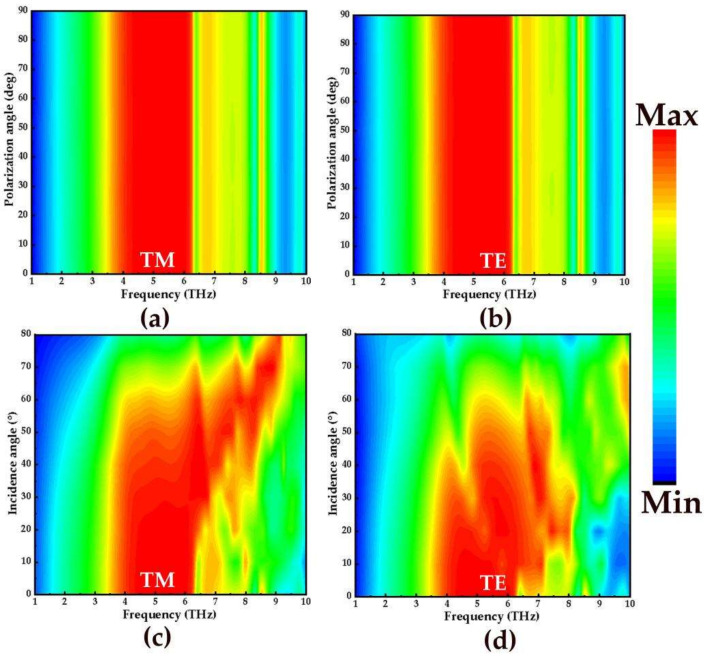
The absorption spectra of broadband absorbers in TM and TE mode: (**a**,**b**) different polarization angles; (**c**,**d**) different incident angles.

**Figure 8 micromachines-14-00201-f008:**
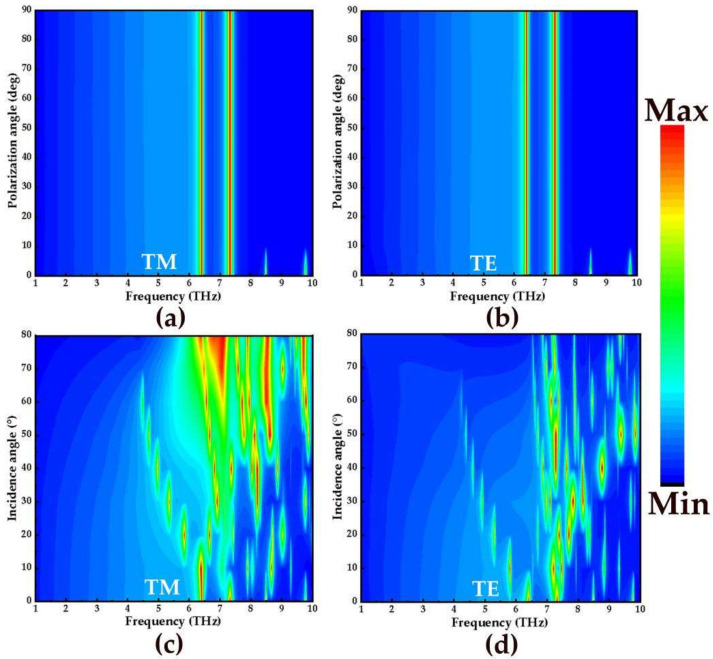
The absorption spectra of narrowband absorbers in TM and TE mode: (**a**,**b**) different polarization angles; (**c**,**d**) different incident angles.

**Table 1 micromachines-14-00201-t001:** Graphene and VO_2_ correspond to the broadband and narrow band performance parameters of the absorber under different conditions.

Fermi Level of Graphene (eV)	Phase Transition of VO_2_	Absorption Bandwidth (THz)	Functionality
0	metallic phase	3.85–6.32	Low-frequency broadband absorption
0	insulating phase	6.4, 7.32, 8.49, 9.78	quadruple narrowband absorption
0.7	metallic phase	6.92–8.92	High-frequency broadband absorption
0.7	insulating phase	\	Single fold narrowband absorption

**Table 2 micromachines-14-00201-t002:** Comparison of the broadband absorption performance between different absorbers.

Reference	Number of Layers	Absorption Bandwidth (THz)	Absorption Amplitude (%)	Tunable Range (%)	Functionality	Polarization Insensitive
[21]	7	0.8–2.4	90	20–95	Broadband and narrowband absorption	Yes
[22]	6	1.05–2.35	90	5.2–90	Dual-broadband absorption	Yes
[23]	3	5.2–6.3	90	None	Broadband and narrowband absorption	Yes
[24]	5	1.05–2.35	90	45.5–90	Broadband and narrowband absorption	Yes
[25]	3	1–2.03	90	25–99.3	Broadband absorption	No
This work	4	3.85–6.32	90	None	Broadband and narrowband absorption, Dual-broadband absorption	Yes

**Table 3 micromachines-14-00201-t003:** Comparison of the narrowband absorption performance between different absorbers.

Reference	Number of Layers	Absorption Bandwidth	Absorption Amplitude (%)	Functionality	Polarization Insensitive
[21]	7	0.7, 2.1, 3.9 THz	100, 100,100	Triple narrowband absorption	Yes
[26]	6	771 nm	99.90	Single narrowband absorption	Yes
[27]	6	0.6, 1.6, 2.8, 3.9, 5.2, 6.3, 7.4, 8.5, 9.6 THz	90, 90, 90, 90, 90, 90, 90, 90, 90	Nine narrowband absorption	Yes
This work	4	6.4, 7.32, 8.49, 9.78 THz	95.2, 97, 89.1, 73.3	Four narrowband absorption	Yes

## Data Availability

The data that support the findings of this study are available from the corresponding author upon reasonable request.

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
