# Peer review of "Tunable Broadband-Narrowband and Dual-Broadband Terahertz Absorber Based on a Hybrid Metamaterial Vanadium Dioxide and Graphene"

_micromachines, 2023, doi:10.3390/mi14010201_

Round 1
Reviewer 1 Report
By using vanadium oxide and graphene at the same time, the absorption of proposed structure in terahertz band can be adjusted. In fact, it is not a new thing to realize broadband narrowband adjustability by using the two materials at the same time. For example, this article https://doi.org/10.1364/OE.391891. The innovation of this work is to propose a new structure to achieve adjustable absorption in different frequency bands. In general, this work lacks innovation, and this article needs further revision. I suggest major revision.
The detailed comments are as follows:
1: The abstract is too long and is hard to extract key points.
2: In fact, the author did not summarize the common problems of existing THz dynamically tunable metasurfaces and the reasons for these problems. This leads to the innovation of the article is not well demonstrated. In fact, in the following comparison table, it is difficult for me to distinguish the advantages of this structure over other structures from one of the parameters.
3: The author has assumed a material with a dielectric constant of 1.96, but is there such a material? Please explain why you chose this material. Frankly speaking, I think this structure lacks the potential of practical application.
4: The resolution of the image is too low, resulting in poor reading experience. In addition, Figure 2 should adopt different line types for data with different parameters.
5: In this paper, under what conditions graphene and vanadium oxide correspond to the broadband and narrowband performance parameters of the absorber are not clear. It can be classified in the form of tables.
6: It is suggested that the author reorganize the full text logic again, especially the introduction.
Reviewer 2 Report
The authors proposed a functionally tunable terahertz (THz) metamaterial absorber based on vanadium dioxide (VO2) and graphene and investigated the absorption characteristics and the tuning of the absorber with simulations. However, some important information is missing to help understand the importance of the work:
(1) The advantage of utilizing such metamaterial structure instead of pure graphene or graphene + vanadium dioxide should be included before the further analysis of the proposed structure. For example, the absorption spectra when there is only a layer of graphene vs graphene + vanadium dioxide thin film. The comparison of the absorption band in graphene + vanadium dioxide thin film when vanadium dioxide is at insulating state vs metallic state is also advised to be included.
TE and TM mode analysis in Figure 3(b) and (c) is not informative. Both VO2 and graphene are simulated as homogeneous in x and y direction and the metamaterial structure is polarization-independent, so the responses for TE and TM excitation are the same for sure. Therefore, I suggest the removal of Figure 3(b) and (c) and could only use one sentence to suggest the same response with TE and TM excitation in the manuscript.
When the vanadium dioxide is tuned from the insulating state to the metallic state, the broadband absorption is actually occurred below the resonant frequencies (<6 THz) and the absorption capability is worse above 6 THz at those resonant frequencies. Why is that? Does that mean the resonances for the structure are actually not that significant to contribute to the broadband absorption? Further explanations should be included to help understand the reason of such structure.
Figure 4 is poorly formed. First, the colors do not show high contrast so the modes inside are hard to interpret. Second, color bars are missing, so it will bring misunderstanding on where the electric field is actually focused. Third, it is better to add “THz” after the numbers inside the figures.
The authors plotted the effective impedance when the conductivity of the vanadium dioxide is changed, but no explanation of why the broadband absorption is formed with the analysis of the effective impedance can be found.
Figure 6 shows the absorption band with different graphene Fermi levels when the vanadium dioxide is at metallic state. Does the Fermi level of graphene change with different temperatures? The authors might need to add some sentences explaining the effect of temperatures on the Fermi levels of graphene, especially under the temperatures when the vanadium dioxide is experiencing the IMT (insulation-to-metal transition).
In Figure 7(c), there is an increased absorption band when the incident angle is increased to around 55 degrees, suggesting better broadband absorption, why is that? Does that mean there would be a better design for which the absorption band could be further broaden at vertical incidence? The authors should include the explanations.
The language of the manuscript and the figure captions should be improved. All figure captions should be further improved to contain more detailed information to avoid misunderstanding. For example, caption of Figure 3(a) should include what is the condition under the broadband absorption and narrowband absorption and could be written as “(a) Curve of broadband absorption (red line) when VO2 is at insulating state and curve of narrowband absorption (blue line) when VO2 is at metallic state”. Although such descriptions can be found in the manuscript, it is better to also included those in the captions.
Reviewer 3 Report
Authors proposed a functionally tunable terahertz metamaterial absorber based on vanadium dioxide and graphene, and investigated the absorption characteristics and the tuning of the absorber by using Computer Simulation Technology microwave studio. I think, paper is interesting, I would propose some changes as follows:
1. Authors should increase the font size in Figures 2, 3, 5, 6.
2. It would be highly desirable to compare the obtained results with the experimental outputs.
3. Authors should clearly describe origin of the absorption in the manuscript.
4. The English language is poor and should be improved.
5. Authors should stress novelty of their work.
6. Authors are missing some recent articles in the field such as Investigation of Hyperbolic Metamaterials, etc.
Round 2
Reviewer 1 Report
The article has been revised to meet the publication requirements.
Reviewer 2 Report
I suggest it to be published on Micromachines with the current version.